# Learning Embodied Vision-Language Programming from Instruction, Exploration, and Environmental Feedback

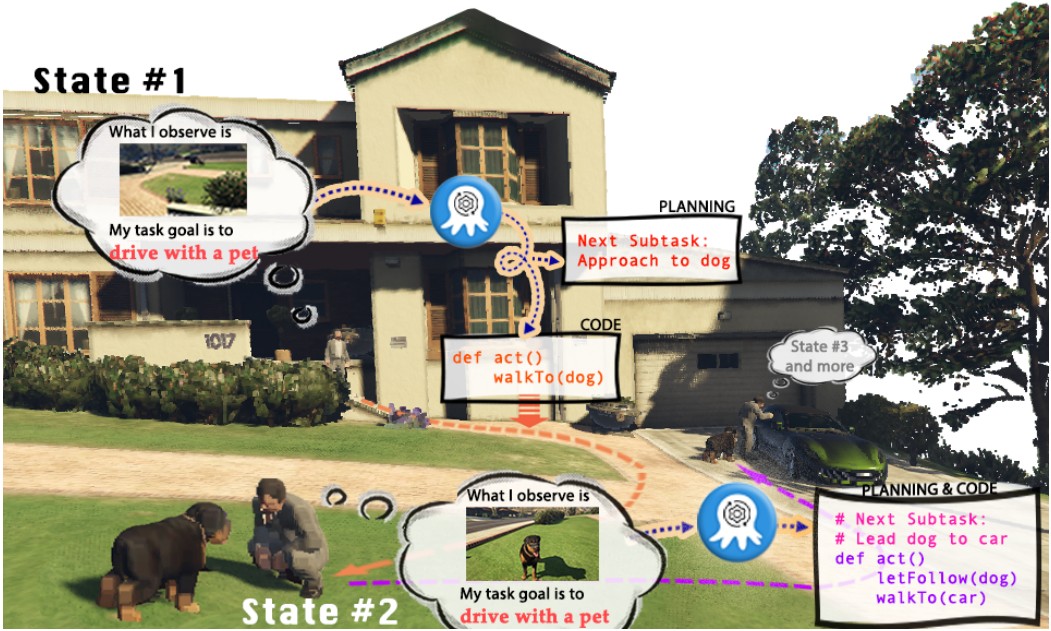

Figure 1: **Illustration of the functionality of our vision-language programmer, Octopus, in the developed OctoGTA environment.** Given a task in the form of natural language, Octopus relies on its egocentric vision to generate plans and the corresponding executable code.

## ABSTRACT

Large vision-language models (VLMs) have achieved substantial progress in multimodal perception and reasoning. Furthermore, when seamlessly integrated into an embodied agent, it signifies a crucial stride towards the creation of autonomous and context-aware systems capable of formulating plans and executing commands with precision. In this paper, we introduce **Octopus**, an embodied VLM designed to **1)** proficiently decipher an agent's visual and textual task objectives, **2)** formulate intricate action sequences, and **3)** generate executable code. Our design allows the agent to adeptly handle a wide spectrum of tasks, ranging from mundane daily chores in simulators to sophisticated interactions in complex video games. Octopus is trained by leveraging GPT-4 to control an explorative agent to generate training data, i.e., action blueprints and the corresponding executable code, within our experimental environment called **OctoVerse**. We also collect the feedback that allows the enhanced training scheme of **Reinforcement Learning with Environmental Feedback (RLEF)**. Through a series of experiments, we illuminate Octopus's functionality and present compelling results, and the proposed RLEF turns out to refine the agent's decision-making. By open-sourcing our model architecture, simulator, and dataset, we aspire to ignite further innovation and foster collaborative applications within the broader embodied AI community.

# 1 INTRODUCTION

With the rise of large language models (LLMs) (Radford et al., 2019; Brown et al., 2020; Ouyang et al., 2022; Touvron et al., 2023; Chiang et al., 2023), a subsequent surge in vision-language models (VLMs) emerged (Alayrac et al., 2022; Awadalla et al., 2023; Li et al., 2023d;b). This evolution has expanded machine capabilities, enabling tasks such as accurate image or video-based descriptions (Li et al., 2023d), reasoning (Xie et al., 2023; Chen et al., 2023), and conversations (Dai et al., 2023; Li et al., 2023b). In the realm of embodied AI, notable efforts like SayCan (Ahn et al., 2022), Palm-E (Driess et al., 2023), and RT-2 (Brohan et al., 2023) have trained on robot manipulation data, so that the agents process visual input and relay precise robotic motor control commands.

Parallel to this robot manipulation approach, another methodology to interact with the environment focuses on task execution through code invocations. This paradigm mirrors our inherent human System-I stimuli, characterized by instinctive actions akin to predefined code. Conversely, the more contemplative System-II processes, which involve planning and reasoning, may be better suited for large models. For example, referring to Figure 1, planning a car ride with a pet might entail a subconscious checklist: `getOutOf()` the house, `check()` for the pet outside, `approach()` the pet, `letFollow()`, and then `open()` to `moveIn()` to the car. In fact, such a "programmatic" paradigm has been, although not in vision, leveraged by pioneering works such as Tool-Former (Schick et al., 2023), HuggingGPT (Shen et al., 2023), ViperGPT (Surís et al., 2023), and VisProg (Gupta & Kembhavi, 2023). They harness LLMs to craft programs and trigger relevant APIs. Game-centric models like Voyager (Wang et al., 2023) and Smallville (Park et al., 2023) have similarly employed GPT for function calls within game engines, though they often parse data directly from their environments.

However, similar programming paradigms are unexplored when incorporating visual perception. Primary initiatives like TAPA (Wu et al., 2023) and SayPlan (Rana et al., 2023) can only output plans, which anchor their strategies in initial environmental states or employ dynamic scene graphs for LLM inputs, respectively. Despite their innovations, the seamless conversion of detailed plans into real-world actions is still missing. Another significant challenge is the over-reliance on pre-trained vision models to convert vision content into language, which can occasionally hinder the LLM's performance. While EmbodiedGPT (Mu et al., 2023) addresses the problem by integrating vision-language modeling for planning and then transitioning to manipulation using policy mapping, the capability of embodied vision-language models to devise executable programs is still largely uncharted territory.

This gap inspired our exploration. In this paper, we introduce **Octopus**, a novel embodied vision-language programmer. Figure 1 illustrates how this model integrates an agent's visual perspective with textual task objectives to devise precise action sequences and yield executable code.

To empower Octopus with its vision-centric programming capabilities, we leveraged GPT-4 to collect training data within our experimental realm, the **OctoVerse**. Here, GPT-4 was provided with intricate system messages, extensive environmental cues, and clearly defined objectives. Based on this input, GPT-4 formulated crucial action strategies and their associated code. Meanwhile, the agent operating in the OctoVerse captured its visual perspectives. Using collected data, Octopus stands out in generating code that seamlessly melds vision, language instruction, and action code.

During the data collection phase, the agent, guided by GPT-4, concurrently receives feedback from simulators about the efficacy of each executed code step, discerning successful moves from unsuccessful ones. This led us to incorporate the **Reinforcement Learning with Environmental Feedback (RLEF)** approach into our pipeline. Successful steps earn rewards, which are then used to train a reward model. Leveraging these insights, we further fine-tune Octopus using Proximal Policy Optimization (PPO) (Schulman et al., 2017). This approach serves as a navigational beacon, sharpening the model's decision-making accuracy."

Empirically, the proposed Octopus model showcases its adaptability and prowess in numerous testing scenarios, yielding promising results on not only routine tasks but also those that need reasoning capabilities. When pitted against existing models, Octopus emerges superior in task planning, code generation, and task execution, with its performance being notably enhanced after the RLEF integration. In sum, our key contributions include:

Table 1: **Overview of Related Embodied AI Models.** The proposed Octopus distinguishes itself from other models as a unified vision-language model for both plan and code generation.

| Models | Release Date | Supported Environment | Vision Model | Code Generator | Action w/ Feedback | LLM Training Enabled |
|---|---|---|---|---|---|---|
| Text2Motion (Lin et al., 2023) | Mar. 2023 | Sim | ✗ | ✓ | ✓ | ✗ |
| Instruct2Act (Huang et al., 2023a) | May 2023 | Sim | ✗ | ✓ | ✗ | ✗ |
| Lang2Rewards (Yu et al., 2023) | Jun. 2023 | Sim | ✗ | ✓ | ✓ | ✗ |
| VoxPoser (Huang et al., 2023b) | Jul. 2023 | Sim | ✓ | ✗ | ✗ | ✗ |
| SayCan (Ahn et al., 2022) | Apr. 2022 | Real | ✓ | ✗ | ✓ | ✗ |
| PALM-E (Driess et al., 2023) | Mar. 2023 | Sim, Real | ✓ | ✗ | ✓ | ✓ |
| RT-2 (Brohan et al., 2023) | Jul. 2023 | Real | ✓ | ✗ | ✓ | ✓ |
| SayPlan (Rana et al., 2023) | Jun. 2023 | Real | ✗ | ✗ | ✓ | ✗ |
| EmbodiedGPT (Mu et al., 2023) | May 2023 | Sim | ✓ | ✗ | ✓ | ✓ |
| TaPA (Wu et al., 2023) | Jul. 2023 | Sim | ✗ | ✗ | ✗ | ✓ |
| Voyager (Wang et al., 2023) | May 2023 | Game | ✗ | ✓ | ✓ | ✗ |
| Octopus | Oct. 2023 | Sim, Game | ✓ | ✓ | ✓ | ✓ |

- A novel embodied vision-language planner and programmer trained with Reinforcement Learning with Environmental Feedback (RLEF).
- Two diverse and realistic embodied environments within the OctoVerse framework: (i) OctoGibson, which is developed upon OmniGibson (Li et al., 2023c), and (ii) OctoGTA, which is adapted from GTA-V (gta, 2014). Based on their platforms, we carefully design tasks and programming function libraries for Octopus.
- Compelling results demonstrating the effectiveness of the integrated RLEF approach in Octopus and useful insights facilitating future research on visual planning and programming.

## 2 RELATED WORK

### 2.1 EMBODIED AI WITH LARGE MODELS

The recent wave of research focuses on merging LLMs with embodied AI tasks (Radford et al., 2019; Brown et al., 2020; Ouyang et al., 2022; Touvron et al., 2023). For instance, VoxPoser addresses robotic manipulation problems through unsupervised methods (Huang et al., 2023b). A group of projects, namely SayCan (Ahn et al., 2022), Palm-E (Driess et al., 2023), RT-2 (Brohan et al., 2023), and EmbodiedGPT (Mu et al., 2023), effectively integrate visual or linguistic cues with robot manipulation data. Outside the domain of robotic manipulation, initiatives like Voyager (Wang et al., 2023) and Smallville (Park et al., 2023) harness the capabilities of GPT to interface with game functions, relying on preset functions to manage intricate manipulations. In a parallel vein, VisProg (Gupta & Kembhavi, 2023) leverages GPT-3 language prompts to craft Python programs, opening the door to a multitude of fascinating applications. While the proposed Octopus model also formulates plans and code, its distinguishing feature is the seamless integration of visual input in program and code generation. This also stands in contrast to other embodied planners like TAPA (Wu et al., 2023) and SayPlan (Rana et al., 2023), which deploy separate vision modules to translate visual data into linguistic inputs for LLMs. Octopus excels as a cohesive vision-language model, delivering not just plans but also executable code.

### 2.2 VISION-LANGUAGE MODELS

Recent advances in large language models (LLMs) like GPTs (Radford et al., 2019; Brown et al., 2020; Ouyang et al., 2022), LLaMA (Touvron et al., 2023), and Vicuna (Chiang et al., 2023) have bolstered the performance of vision-language models, such as Flamingo (Alayrac et al., 2022; Awadalla et al., 2023) and BLIP-2 (Li et al., 2023d), particularly in zero-shot learning scenarios. To advance the conversation and interaction capabilities of vision-language models, researchers have begun exploring more. These include Otter (Li et al., 2023b), InstructBLIP (Dai et al., 2023), and LLaVA (Liu et al., 2023), among other noteworthy contributions (Ye et al., 2023; Zhou et al., 2022a; Li et al., 2023a). These models are specifically designed to facilitate complex human-model interactions and are particularly well-suited for use in multi-modal chatbots. Extended from Otter (Li et al., 2023b), we propose Octopus, the vision-language programming model designed to facilitate

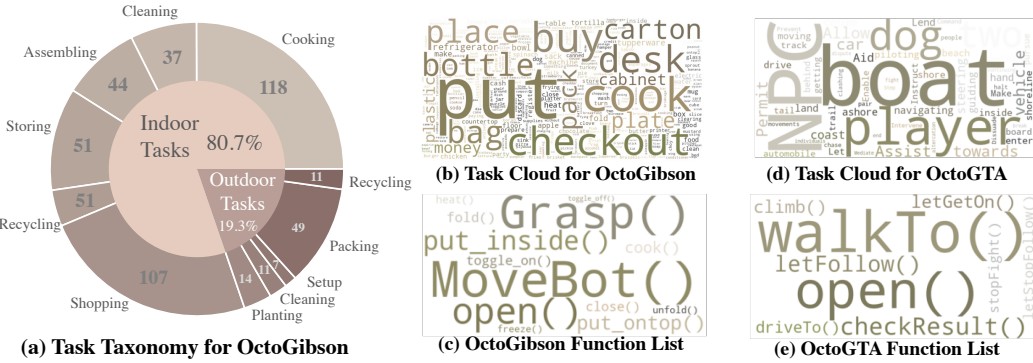

Figure 2: **The Statistics of the OctoVerse Environment.** We present the task composition and the function word cloud for both simulator environments.

**human-model-agent** interaction. Specifically, Octopus processes human instructions to generate action codes, enabling agents to execute operations accordingly.

## 2.3 FEEDBACK IN LARGE LANGUAGE MODELS

Reinforcement Learning with Human Feedback (RLHF) (Ouyang et al., 2022; Stiennon et al., 2020; Ziegler et al., 2019) is a modern approach in the field of AI that combines traditional reinforcement learning with feedback from human supervisors. (Sun et al., 2023) is the first successful adaptation of RLHF to vision-language alignment. In our research, we propose Reinforcement Learning with Environmental Feedback (RLEF), which harnesses the power of environmental feedback to train an embodied vision-language model. Instead of direct human supervision, the feedback in RLEF naturally comes from the simulator environment.

## 3 THE OCTOVERSE ENVIRONMENT AND DATA COLLECTION

In this section, we present the simulator environments designed to train and assess the Octopus model. We then delve into our data collection techniques utilized within these environments and explain the detailed information of the data in training and test sets.

### 3.1 OVERVIEW OF OCTOVERSE

To train our Octopus model, we developed two simulator environments under the unified name of OctoVerse. Our primary environment is the OctoGibson simulator, from which we collect the training data and conduct our primary analysis. We then assess the model's generalization capabilities in the OctoGTA simulator.

**OctoGibson** We built the environment on the foundation of an existing simulation framework, OmniGibson (Li et al., 2023c), which supports 1,000 daily activities across 50 scenes, featuring over 5,000 meticulously annotated objects. To bolster model training, we incorporated **16 functions** that the robot can execute, such as `walkTo()`. Within this environment, we meticulously crafted **476 tasks**[1]. Each task begins with an initial state and concludes with a definitive termination state, allowing for a straightforward assessment of task completion. Among them, 367 tasks are **routine tasks**—simple and direct actions like "place a glass in a trash can". Conversely, the remaining 109 are **reasoning tasks** which necessitate deeper comprehension. An example is "buy a chocolate", where the agent needs to know to pick a chocolate bar from the shelf and then place it, along with money, on the checkout counter. To acquaint readers with our environment, Figure 2 (a-c) illustrates the task taxonomy and provides a word cloud.

**OctoGTA** Our secondary environment, built on the foundation of GTA-V (gta, 2014), serves the purpose of auxiliary experiments, assessing the Octopus model's generalizability. Within this set-

---

[1]The full list of task names and their categories are listed in this google sheet.

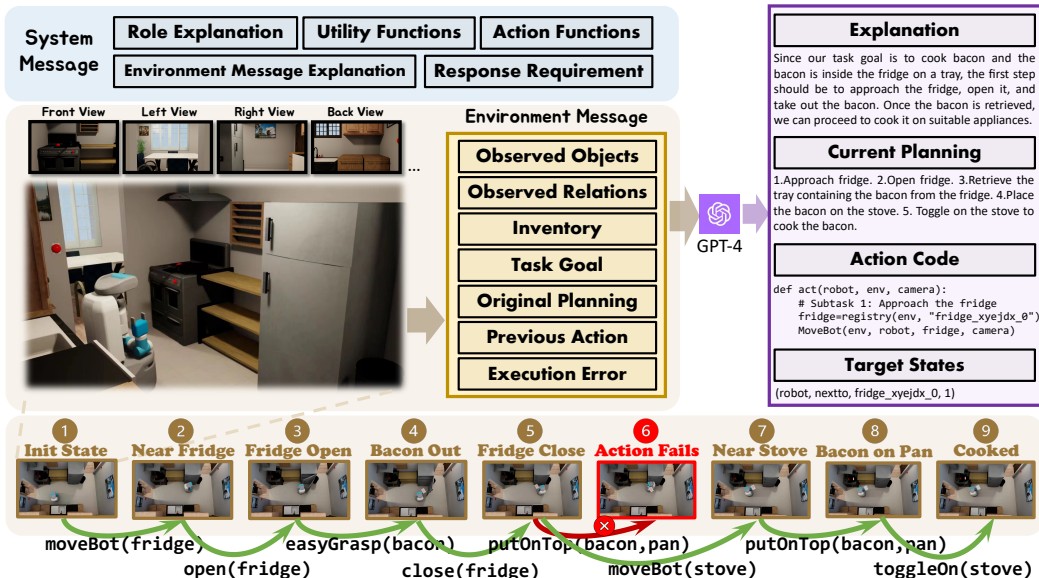

Figure 3: **Data Collection Example for "Cook a Bacon" Task.** GPT-4 perceives the environment through the `environmental message` and produces anticipated plans and code in accordance with the detailed `system message`. This code is subsequently executed in the simulator, directing the agent to the subsequent state. For each state, we gather the environmental message, wherein `observed objects` and `relations` are substituted by egocentric images to serve as the training input. The response from GPT-4 acts as the training output. Environmental feedback, specifically the determination of whether each target state is met, is documented for RLEF training.

ting, we've integrated **11 functions** and methodically crafted **20 tasks**[2]. Apart from the example in Figure 1, another example of such a task is "help NPC to drive their boat back to shore".

## 3.2 INSTRUCTIONS FROM EXPLORATION

Initiating the training of the Octopus model involves ensuring its operational capability, particularly its ability to process vision input, interpret current and past states (such as objects the agent is holding), and produce structured plans and executable code. Thus, the primary task in organizing training data is to form a succinct pairing: "vision input + current/historical states → next step plan + executable code". However, collecting these pairs is far from simple; manually pairing them through human programmers would be both time-intensive and laborious. To circumvent this challenge, we harness the capabilities of GPT-4, not only to guide the agent's actions for task attempts but also to facilitate the automated data-gathering process.

**Environment Info Collection** As delineated in Figure 3 and Figure 4 (a), we harvest an **environment message** for each state, encompassing attributes like `Observed Objects`, `Observed Relations`, `Inventory`, and more. Specifically, the simulator can provide us with an exact scene graph at each state, shaping the content for the first two parts. The inventory info can be easily obtained in the simulator. The task, e.g., "cooking bacon" in Figure 3, is represented by the `Task Goal`.

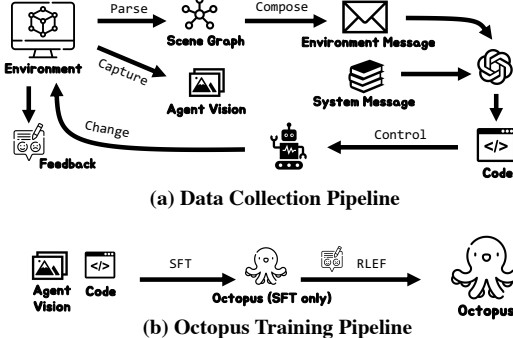

(a) Data Collection Pipeline

(b) Octopus Training Pipeline

Figure 4: Data Collection and Training Pipeline

**Automation with GPT-4** Having prepared the environment message, we next crafted a structured **system message** to ensure that the robot not only understands its input but also maintains a consistent output format. A detailed examination of this prompt can be found in the appendix. Experiments have shown that a well-articulated prompt

---

[2]We meticulously design tasks to be friendly, ensuring they exclude any inappropriate or violent behaviors.

allows GPT-4 to effectively generate executable codes. It's worth noting that the combined length of the system and environment messages can be extremely long. As a result, standard GPT-4 8K models may struggle to produce meaningful outputs, necessitating the use of the more robust GPT-4 32K model. As illustrated in Figure 3, when GPT-4 is fed a consistent system and environment message, it yields comprehensive outputs, encompassing current scenario analysis, planning, and actionable codes. The data will support the training in Section 4.2.

**Error Management** Notably, GPT-4 collects training data under the main task of guiding the agent to complete tasks. However, GPT-4 is not infallible. Errors can manifest in multiple ways, ranging from syntax errors to physical challenges in the simulator. For instance, as depicted in Figure 3, between states #5 and #6, the action failed due to the long distance between the agent (bacon) and pan. Such setbacks reset the task to its previous state. If a task remains incomplete after 10 steps, it is deemed unsuccessful, and we terminate this task for budget concerns. All data pairs, regardless of the task's completion status, are valuable resources for refining instructions.

## 3.3 Environmental Feedback

While GPT-4 guides the agent toward task completion, its continual trial-and-error approach does more than just collect vision-output pairs. This iterative problem-solving provides a rich set of feedback data. The automatic annotation of the feedback is twofold, focusing on both step-level and task-level judgments. **Step-level judgment** assesses the alignment of post-execution states with their target states. For instance, in Figure 3, steps color-coded in green signify positive feedback. One can visualize the action sequence for task completion as a tree, where each node indicates a step (subtask), encapsulating an action code. Accompanying each step is a binary value that denotes success or failure, giving preference to the successful branch over its counterpart. **Task-level judgment**, on the other hand, gauges the successful execution of the overall task. If the task is not completed as intended, every state within that task is labeled as negative. This collated feedback data serves as a foundation for our Reinforcement Learning with Environmental Feedback (RLEF) methodology, which we discuss in greater detail in Section 4.3.

## 4 Octopus: The Embodied Vision-Language Programmer

In this section, we delineate the architecture and training methodologies underpinning **Octopus**, our novel vision-language programmer. Building upon the foundational principles of Otter (Li et al., 2023b), Octopus incorporates specialized modules to cater to the vision-language programming tasks within OctoVerse. We will elucidate the architectural design derived from the Otter model, detail the supervised fine-tuning approach that harnesses instructions from exploration, and explore the integration of reinforcement learning enhanced by environmental feedback. We refer to Figure 4 (b) which briefly illustrates the Octopus training pipeline.

### 4.1 Architecture

The Octopus architecture is heavily inspired by the foundation laid by the Otter model (Li et al., 2023b). However, in our adaptation, specialized modifications have been made to tailor the architecture for the unique challenges of vision-language programming tasks found in OctoVerse. At the core of Octopus is the seamless integration of two critical components: **MPT-7B Language Decoder** (MosaicML, 2023) and **CLIP VIT-L/14 Vision Encoder** (Radford et al., 2021).

To further enhance the synergy between the vision and language components, we have incorporated design principles from the Flamingo architecture (Alayrac et al., 2022). This is evident in our employment of the **Perceiver Resampler module** and the intricate weaving of **Cross-Gated Attention modules**. Initially, the Perceiver Resampler module ingests a sequence of image or video features to produce a fixed set of visual tokens. Subsequently, these tokens condition the language layers through Cross-Gated Attention modules, where the tokens act as keys and values while text from preceding layers serves as queries.

Through this detailed architecture, the Octopus is primed to excel in tasks that demand a nuanced understanding of both visual and textual data.

## 4.2 SUPERVISED FINETUNING WITH INSTRUCTIONS FROM EXPLORATION

We train the Octopus model on our collected dataset from OctoVerse $\mathcal{D}_{\mathrm{E}} = \{(\mathbf{X}_v, \mathbf{T}_i, \mathbf{T}_r)\}$ with token-level supervised fine-tuning (SFT) (Ouyang et al., 2022; Touvron et al., 2023). During training, the Perceiver Resampler transforms images $\mathbf{X}_v$ into visual tokens that are aligned with text modality in the language model layers. These visual tokens condition subsequent layers via Cross-Gated Attention modules. The training objective involves next-token prediction, akin to GPT series models (Brown et al., 2020; OpenAI, 2023), additionally with the incorporation of visual and textual inputs. The likelihood of a targeted response $\mathbf{T}_r$ is modeled as follows:

$$p(\mathbf{T}_r \mid \mathbf{T}_i, \mathbf{X}_v) = \prod_{l=1}^{L} p(t_l \mid \mathbf{X}_v, \mathbf{T}_i, \mathbf{T}_{r,<l}). \tag{1}$$

Note that $\mathbf{T}_i$ denotes the instruction tokens and $\mathbf{T}_{r,<l}$ denotes the response tokens before the current predicted token $t_l$. During inference, tokens are converted into natural language via the language decoder's text tokenizer.

In OctoVerse, visual observations are represented by $\mathbf{X}_v = \{x_F^0, \ldots, x_F^7, x_B^0, x_B^1\}$, consisting of eight first-person view (FPV) images followed by two bird's-eye view (BEV) images. During training, this multi-image input $\mathbf{X}_v$ is treated as a continuous video frame sequence. The rationale behind capturing both FPV and BEV is twofold. Firstly, by capturing the FPV, we aim for the agent to mimic human-like processing, assimilating images it directly observes, much like how humans interpret their immediate surroundings. Secondly, the BEV is integrated because agents, unlike humans, can tap into alternative camera sources, such as surveillance cameras, granting a more holistic understanding of the environment. To obtain the eight FPV images, we capture one image every 45 degrees, ensuring a complete 360-degree perspective of the environment.

## 4.3 REINFORCEMENT LEARNING WITH ENVIRONMENTAL FEEDBACK (RLEF)

Within the OctoVerse ecosystem, as explained in Section 3.3 and Figure 3, we visualize task progression as a tree. Each node on this tree symbolizes a sub-task, and it carries a binary value, either $\{0, 1\}$, to denote if the sub-task was successful or not. Simply put, if a node (or sub-task) has a value of 1, it is a step in the right direction toward our end goal.

**Tree-based Task Representation**   We organize these data into environmental reward datasets $\mathcal{D}_{\mathrm{R}} = \{(\mathbf{X}_v^*, \mathbf{T}_i^*, \mathbf{T}_r^i, \mathbf{T}_r^j, c)\}$ where $\mathbf{T}_r^i$ and $\mathbf{T}_r^j$ are two responses on the tree with the same parental node's task description $\mathbf{T}_i^*$, and $c$ is the index of preferred response that could lead to final completion of the given task. The primary purpose of this step is to ensure that, when faced with two sub-tasks stemming from the same parent task, the reward mechanism favors the branch that is successfully executed. Note that even if a parental node does not have multiple responses, we can still assign feedback according to Section 3.3.

**Reward Model Configuration**   We finetune a single-modal CodeLLaMA-7B model on $\mathcal{D}_{\mathrm{R}}$ with an additional value head as our reward model $r_\phi$. For computational efficiency, the reward model is designed to accept only textual modality and outputs a scalar reward. The function of this text-based reward model is to assess state transitions, denoted by $\mathbf{T}_i^* \to \mathbf{T}_r^{i,j}$, to determine which transitions yield higher rewards and thereby assist the agent in task execution and completion.

**Policy Model Development**   Next, we employ the above supervised fine-tuned model as the initial policy model (Ouyang et al., 2022) $\pi^{\mathrm{INIT}}$ with fixed parameters. Then we initialize another duplicate of the model as the RL-tuned model $\pi_\theta^{\mathrm{RL}}$, and train it with Proximal Policy Optimization (PPO) (Schulman et al., 2017) to maximize response rewards. The loss function is formulated as:

$$\mathcal{L}\left(\pi_\theta^{\mathrm{RL}}\right) = -\mathbb{E}_{(\mathbf{X}_v^*, \mathbf{T}_i^*) \in \mathcal{D}_{\mathrm{R}}, \mathbf{T}_r \sim \pi^{\mathrm{RL}}} \left[ r_\phi(\mathbf{T}_i^*, \mathbf{T}_r) - \beta \cdot \mathbb{D}_{\mathrm{KL}} \left( \pi_\theta^{\mathrm{RL}}(\mathbf{X}_v^*, \mathbf{T}_i^*) \parallel \pi^{\mathrm{INIT}}(\mathbf{X}_v^*, \mathbf{T}_i^*) \right) \right],$$
$$\tag{2}$$

where $\beta$ acts as a hyper-parameter to regulate the magnitude of the Kullback–Leibler (KL) penalty.

## 5 EXPERIMENTS

**Experimental Setup**   We first set up the OctoGibson to evaluate the performance of Octopus and other related models. Specifically, we are utilizing the metrics of goal task completion score to check

Table 2: **Main Results on OctoGibson.** We compare various models: standalone language models, adapted vision-language planners, and our Octopus models, across different evaluation settings. In cells displaying two values, the first represents the task completion rate across the target validation task sets, while the second assesses the conceptual accuracy of the model's planning as judged by human evaluators. GT denotes that the model input is directly parsed from the simulator, with information on objects (O) or relations (R). Octopus shows consistently better results on task completion.

| Model | Vision Model | Language Model | Entire Goal Task | | | | |
|---|---|---|---|---|---|---|---|
| | | | Seen Env | Unseen Env | Follow | Reason | All |
| LLaMA | GT (O+R) | LLaMA2-7B | 0.07 / 0.11 | 0.13 / 0.13 | 0.11 / 0.16 | 0.00 / 0.00 | 0.08 / 0.12 |
| CodeLLaMA | GT (O+R) | CodeLLaMA-7B | 0.09 / 0.20 | 0.20 / 0.40 | 0.16 / 0.31 | 0.00 / 0.07 | 0.12 / 0.25 |
| TAPA (task-level) | ~~OVD~~ GT (O) | CodeLLaMA-7B | 0.09 / 0.36 | 0.13 / 0.33 | 0.11 / 0.36 | 0.06 / 0.33 | 0.10 / 0.35 |
| TAPA (step-level) | ~~OVD~~ GT (O) | CodeLLaMA-7B | **0.16** / **0.42** | 0.13 / 0.27 | **0.18** / 0.38 | 0.07 / 0.40 | 0.15 / 0.38 |
| EmbodiedGPT | CLIP-ViT | MPT-7B | 0.04 / 0.36 | 0.27 / **0.53** | 0.13 / 0.38 | 0.00 / 0.40 | 0.10 / 0.40 |
| Octopus (SFT Only) | CLIP-ViT | MPT-7B | 0.11 / 0.33 | 0.27 / 0.47 | 0.16 / 0.38 | 0.13 / 0.33 | 0.15 / 0.37 |
| Octopus (SFT + RLEF) | CLIP-ViT | MPT-7B | 0.13 / 0.38 | **0.33** / **0.53** | **0.18** / **0.40** | **0.20** / **0.53** | **0.18** / **0.42** |

whether the task is actually completed in the simulator and the plan score from human evaluation. We totally have 60 evaluation tasks, with 45 from the seen environment, and 15 that are unseen during training. We also have 45 routine tasks and 15 require reasoning. Please note that models like Octopus might not always accurately identify specific object names as they appear in the simulator (e.g., "water_bottle_189"). To address this, we implement a post-processing step for the generated code, substituting generic object references with their exact names from the simulator with simple string similarity matching.

## 5.1 MAIN RESULTS

**CodeLLaMA Improves Coding but not Planning.** The first two rows in Table 2 highlight the suboptimal task completion rate of the blind LLMs. Among them, CodeLLaMA boasts pre-training on a large programming dataset, resulting in a notable enhancement in code execution from our observation, with 92% of the written code being successfully executed compared to LLaMA's 24%. However, its prowess in planning remains limited. In contrast, the proposed Octopus MPT-7B model displays superior planning and task completion metrics while maintaining commendable coding abilities (72% of the written code can be executed). We surmise that the coding requirements within the OctoGibson environment might not be exceedingly intricate, rendering an advanced programming language model, like CodeLLaMA, less crucial, albeit beneficial. For more insight, although not shown in the table, our efforts to replace the MPT model with CodeLLaMA encountered challenges of generating non-sense outputs, suggesting that more refined code, or image-code paired data might be necessary for a successful Octopus-CodeLLaMA integration.

**Blind LLMs Struggle with Extended Input Content.** Our observations indicate that the step-level TAPA model, when supplied with a ground-truth object list, achieves a notable enhancement in planning. The primary distinction between it and the blind CodeLLaMA lies in the input length; the latter deals with protracted, pairwise relation content, complicating the language model's ability to extract crucial data from the environment message. This scenario highlights the inherent limitation of blind LLMs: relying on language alone to convey the entirety of environmental data can result in unwieldy and less informative input.

**Octopus Demonstrates Superior Task Generalization.** Table 2 underscores Octopus's commendable performance, evidencing its consistent edge over standalone language models in task completion. Its adeptness in adapting to previously unencountered environments underlines the inherent advantages of vision-language models. A more detailed ablation analysis is provided in the subsequent section.

**RLEF Enhances Octopus's Planning Strategy.** Table 2 unequivocally underscores Octopus's profound reasoning capabilities after the RLEF finetuning. An example can be observed in Figure A1(b-c), where, after refinement via RLEF, Octopus astutely navigates to the cabinet housing the carboy instead of attempting a direct yet distant capture. Quantitatively, Octopus exhibits enhanced adaptability to previously unseen reasoning tasks, reinforcing its prowess in logical task resolution. When juxtaposed with other strategies, such as the embodied queries employed by EmbodiedGPT, RLEF emerges as the more efficacious approach.

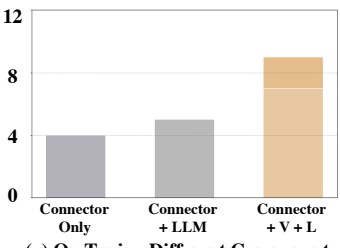 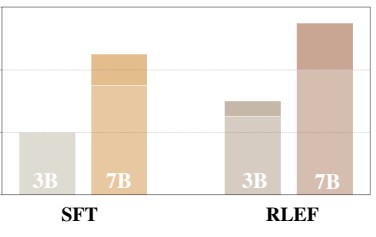 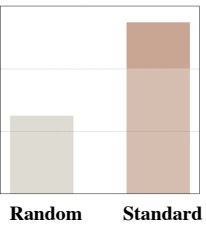

Figure 5: Ablation Study on model components, model size, and vision input. For bars with different colors, the upper bar denotes the number of successful reasoning tasks, and the lower is routine tasks.

## 5.2 ABLATION STUDY

**7B *v.s.* 3B Model Size** We embarked on experiments centered on model size to discern the influence of the total parameter count on the efficacy of vision-language models. As illustrated in Figure 5 (a), downsizing the model manifests in a noticeable performance drop. The congruency of results across both the SFT and RLEF models underscores the importance of an apt model size when sculpting vision-language models.

**Examining Training Components** Through experimentation on training components, we aimed to illuminate optimal strategies for finetuning vision-language models. Figure 5 (b) demonstrates that solely adjusting the connector culminates in success for merely 4 out of 60 tasks. Conversely, finetuning both the connector and language decoder nudges the success rate slightly higher, with 5 tasks being accomplished. In contrast to the fully optimized model, these outcomes accentuate the paramountcy of trained parameters.

**Significance of Visual Inputs in Task Performance** In our standard configuration, the vision component processes a sequence of image inputs, consisting of eight circularly captured first-person view (FPV) images, complemented by two bird's-eye view (BEV) images. With the intent to investigate the impact of visual inputs on task performance, we initiated an ablation study. In a modified setup, the sequence of these visual inputs was deliberately randomized, aiming to attenuate the strength of the visual signals. As illustrated in Figure 5 (c), this intentional disruption in visual input consistency led to a pronounced decline in task performance. This result highlights the crucial role that clear and structured visual inputs play in the Octopus model, emphasizing that it significantly leverages visual cues for effective planning and task execution.

## 5.3 PERFORMANCE OF GPT-4 AND GPT-4V

**Performance of GPT-4** The input provided to GPT-4 was consistent with the input during our data collection phase, which was purely textual. Under such conditions, out of a total of 60 test tasks, GPT-4 achieved a commendable success rate in 31 tasks. This result suggests that current models still possess considerable room for advancement. The fact that even GPT-4 doesn't perform optimally indicates a vast scope for improvements within the domain.

**Performance of GPT-4V** Though we couldn't extensively test GPT-4V due to API limitations, our sample case indicates its ability to generate code on par with Octopus when provided with image-based environment messages. However, while Octopus, having been trained in the present environment, adeptly performs tasks like "open the cabinet", GPT-4V's actions, shown in Figure A1 (e), although seemingly accurate, fall short in specific tasks such as locating the target object - the carboy. Given GPT-4V's zero-shot learning approach and its unfamiliarity with our environment, alongside potential simulator discrepancies, its results remain commendable.

## 5.4 TRANSFERABILITY ON GTA TASKS

To examine Octopus's adaptability in novel environments, we transitioned the model initially trained on OctoGibson to tasks within the GTA framework. We observed that even in a few-shot scenario, Octopus demonstrates commendable performance and can complete 4 out of the 11 test tasks.

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
