# OpenReview forum: "Learning Embodied Vision-Language Programming From Instruction, Exploration, and Environmental Feedback"
_ICLR.cc/2024/Conference — Submitted to ICLR 2024_

### Official Review · Reviewer_VoZ1 · 2023-10-26

**Soundness:** 3 good
**Presentation:** 2 fair
**Contribution:** 3 good
**Rating:** 5
**Confidence:** 4

**Summary:**

The authors propose an embodied vision-language planner and programmer (Octopus) trained with reinforcement learning with environmental feedback, as well as two embodied environments that yield feedback necessary to train the aforementioned model, with data collected by GPT4. Octopus takes egocentric views and tasks specified in language, and outputs next actions and code to execute it. The method is tested on environments based on OmniGibson and GTA-V.

**Strengths:**

I appreciate the introduction of reinforcement learning from environmental feedback, by efficiently using environmental rewards from the simulator + GPT-4. Though the approach of using code-writing LLMs to execute plans is not new, I believe applying it to embodied tasks in the proposed formulation is a nice demonstration of how to leverage foundation models in these embodied environments.

**Weaknesses:**

W1. The data collection process relies on GPT-4, which takes as input language of systems message + environment message to output the required plan, code, and target state. This relies on the strong assumption that the systems + environment message fully captures the environment state, since the planning must be done without access to the view of the visual scene. I presume this means that the systems prompt must be elaborate, handcrafted, and task specific, such that GPT-4 can plan reasonably in the environment using possible objects at hand. Is there a robust way of designing such prompts for different/new tasks without tuning?

W2. From my understanding, GPT-4 generates the target states, and whether the target states have been met is used as environmental feedback for training Octopus. It seems possible that GTP-4 will generate an incorrect or trivial target state to satisfy the language task goal, and be able to successfully reach that predicted target state, without actually achieving the task goal. Is this understanding correct? In this case, the errors from GPT-4 seem more harmful than having unsuccessful execution from GPT-4 generated code.

W3. The results on Table 2 are not particularly convincing of this method’s success. Octopus should indeed outperform blind LLMs that do not take visual input. TAPA outperforms/is of equal performance in 2 of the 5 tasks. The paper lacks analysis on why this is the case. Where does TAPA fail? It is also hard to compare when the vision models are different; does OVD and CLIP-ViT perform similarly in terms of capturing information from the input scene?

Nit: Should add a figure describing Octopus model architecture; notations in methods section are not well defined.

**Questions:**

Q1. Is there a systematic way of generating the systems message able to generalize to new datasets? Or does it need to be hand-crafted and tuned for Omniverse & GTA & other datasets?

Q2. Can you provide more analysis or experiments on where Octopus may outperform prior work?

---

> ### Author Response · Authors · 2023-11-22
> **Response to Reviewer VoZ1(1/2)**
>
> > W1. The data collection process relies on GPT-4, which takes as input language of systems message + environment message to output the required plan, code, and target state. This relies on the strong assumption that the systems + environment message fully captures the environment state, since the planning must be done without access to the view of the visual scene. I presume this means that the systems prompt must be elaborate, handcrafted, and task specific, such that GPT-4 can plan reasonably in the environment using possible objects at hand. Is there a robust way of designing such prompts for different/new tasks without tuning?
>
> We thank the reviewer for your valuable comments. You are right, we make sure the environment message fully captures the environment state since when we design a task, we manually inspect all the necessary objects that should be in the environment message. An automatic alternative is to put everything, no matter whether necessary or not, into the environmental message. However, we tried but we found that GPT-4 will not act properly if the environmental message is too long. However, since OpenAI keeps enlarging the input length support, maybe this issue could be addressed in the future. Besides, since this work is to have an initial exploration of the vision-language programming, which remains uncharted to date, we think some handcrafted work is required, and support our future work of making the training pipeline more flexible, such as iterative training, i.e., with an Octopus model that is trained with limited half-handcrafted environment message, the model itself can explore in the environment with visual input, and collect data for next training loop. However, we will keep this exploration in our future exploration.
>
> > W2. From my understanding, GPT-4 generates the target states, and whether the target states have been met is used as environmental feedback for training Octopus. It seems possible that GTP-4 will generate an incorrect or trivial target state to satisfy the language task goal, and be able to successfully reach that predicted target state, without actually achieving the task goal. Is this understanding correct? In this case, the errors from GPT-4 seem more harmful than having unsuccessful execution from GPT-4 generated code.
>
> Yes, sometimes GTP-4 will generate a trivial target state to satisfy the language task goal, but in this case, the main task will not be met, and according to Task-level judgment in Section 3.3, all the steps will marked as failure once the main task goal is not met, and the RLEF training will use the failure to train the model not to do such trivial plan. However, these trivial steps, as long as they do not make syntax errors, will be used for SFT training, helping the model to write good code.
>
> > W3. The results on Table 2 are not particularly convincing of this method’s success. Octopus should indeed outperform blind LLMs that do not take visual input. TAPA outperforms/is of equal performance in 2 of the 5 tasks. The paper lacks analysis on why this is the case. Where does TAPA fail? It is also hard to compare when the vision models are different; does OVD and CLIP-ViT perform similarly in terms of capturing information from the input scene?
>
> Actually, the input of the blind LLMs in Table 2 is the textual representation of the objects (and the relations) that are shown in the vision image, i.e, we put everything that exists in the robot's vision into the text to feed into LLMs, so although it is blind, it can "listen" what the environment has. So the higher task completion rate of Octopus does show the superiority of vision input and vision components in Octopus. In Table 2, Octopus reaches a good result on almost all the metrics except the seen environment, showing that Octopus, with the vision as the input can have a better generalization ability in planning. The pure language models, like TAPA, when given the familiar textual input that reflects the environment, seem to have a better result. But we believe task and environmental generalization is a more important aspect.
>
> > Nit: Should add a figure describing Octopus model architecture; notations in methods section are not well defined.
>
> Thank you for pointing this! We draw the training pipeline in Figure 4, as the architecture is simply following the Flamingo framework (since Otter follows Flamingo), which can be accessed [here](https://github.com/lucidrains/flamingo-pytorch).

---

> > ### Author Response · Authors · 2023-11-22
> > **Response to Reviewer VoZ1(2/2)**
> >
> > > Q1. Is there a systematic way of generating the systems message able to generalize to new datasets? Or does it need to be hand-crafted and tuned for Omniverse & GTA & other datasets?
> >
> > For the system message, according to Appendix D, we think the stuff about the action list is definitely required to be revised for other environments. Other parts might only need minor revision.
> >
> >
> > > Q2. Can you provide more analysis or experiments on where Octopus may outperform prior work?
> >
> > Yes, we have added several experiments in Section 5.1 and Table 2, where we add throughout comparison with "Blind LLM" including LLaMA, CodeLLaMA, TAPA, which can only "listen" to what is inside the environment. We also implement the EmbodiedGPT for comparison. These experiments have shown several conclusions including:
> > - Blind LLMs Struggle with Extended Input Content.
> > - CodeLLaMA Improves Coding but not Planning.
> > - Octopus Demonstrates Superior Task Generalization Ability as it integrates the visual input.
> > -  RLEF Enhances Octopus’s Planning Strategy, which is more significant than the components in EmbodiedGPT.
> >
> > | Model             | Vision Model | Language Model | Seen Env | Unseen Env | Follow | Reason | All   |
> > |------------------|--------------|----------------|----------|------------|--------|--------|-------|
> > | LLaMA            | GT (O+R)     | LLaMA2-7B      | 0.07 / 0.11 | 0.13 / 0.13 | 0.11 / 0.16 | 0.00 / 0.00 | 0.08 / 0.12 |
> > | CodeLLaMA        | GT (O+R)     | CodeLLaMA-7B   | 0.09 / 0.20 | 0.20 / 0.40 | 0.16 / 0.31 | 0.00 / 0.07 | 0.12 / 0.25 |
> > | TAPA (task-level) | ~~OVD~~ GT (O) | CodeLLaMA-7B   | 0.09 / 0.36 | 0.13 / 0.33 | 0.11 / 0.36 | 0.06 / 0.33 | 0.10 / 0.35 |
> > | TAPA (step-level) | ~~OVD~~ GT (O) | CodeLLaMA-7B   | **0.16 / 0.42** | 0.13 / 0.27 | **0.18 / 0.38** | 0.07 / 0.40 | 0.15 / 0.38 |
> > | EmbodiedGPT       | CLIP-ViT     | MPT-7B         | 0.04 / 0.36 | 0.27 / **0.53** | 0.13 / 0.38 | 0.00 / 0.40 | 0.10 / 0.40 |
> > | Octopus (SFT Only) | CLIP-ViT   | MPT-7B         | 0.11 / 0.33 | 0.27 / 0.47 | 0.16 / 0.38 | 0.13 / 0.33 | 0.15 / 0.37 |
> > | Octopus (SFT + RLEF) | CLIP-ViT | MPT-7B         | 0.13 / 0.38 | **0.33 / 0.53** | **0.18 / 0.40** | **0.20 / 0.53** | **0.18 / 0.42** |
> >
> > These experimental results show the promising direction of the end-to-end vision language models for planning and programming.

---

> > > ### Author Response · Authors · 2023-11-23
> > >
> > > We sincerely appreciate your great efforts in reviewing this paper. Your constructive advice and valuable comments really help improve our paper. Considering the approaching deadline, please, let us know if you have follow-up concerns. We sincerely hope you can consider our reply in your assessment, and we can further address unclear explanations and remaining concerns if any.
> > >
> > > Once more, we are appreciated for the time and effort you've dedicated to our paper.

---

### Official Review · Reviewer_ABA1 · 2023-10-29

**Soundness:** 2 fair
**Presentation:** 1 poor
**Contribution:** 2 fair
**Rating:** 3
**Confidence:** 5

**Summary:**

This paper leverages GPT4 to generate vision and language training data from OmniGibson and GTA-V. Then, based on the data, they train the model modified from the Otter model and perform some downstream embodied tasks to demonstrate the performance.

**Strengths:**

1. The idea of using GPT4 with crafted prompts to acquire training data from existing environments is interesting.

2. Error management and environment feedback are reasonable.

**Weaknesses:**

1. The novelty of this paper is limited. Essentially, it uses GPT4 with the prompt engineer to collect data from two embodied environments and then trains a vision-language agent model. Moreover, The agent model does not have a specific framework diagram to show the detailed parts, making it difficult to see which part of the model is its innovation point. I suggest providing a framework diagram to clearly explain where the model is newly proposed in the paper and how it differs from existing methods.

2. This paper appears to employ an intentional use of uncommon or less frequently used words in many sentences, substituting them for simpler, more common terms that could convey the message more clearly. As a result, the reading experience becomes somewhat disjointed, and the text may come across as rather weird to the reader.

3. The reason for using both OmniGibson and GTA-V environments to generate data seems not obvious. A more obvious comparison between the two environments is required, such as the visual comparison of the tasks.

4. In Section Error Management, when the agent executes the wrong command, how does the method perform "error management" on it? It seems that this section only claims the cases under which a task is defined as failure, and does not show how to manage such failure.

5. In Section ENVIRONMENTAL FEEDBACK, what if there are multiple erroneous states in a task sequence?  Which are the positive states and which are the error states at this time? And it may not be said that if one of the states is wrong, then its previous states must all be negative.

6. The experimental baselines are unfair and unclear. For Blind LLMs, without visual input to GPT4, GPT4 cannot ground language into the visual environment, which will inevitably lead to worse results. As for TAPA, the reader cannot understand what kind of model it is and its workflow. Even the OVD are not introduced or cited.

7. What are the tasks of the four testing environments? The authors did not give a detailed introduction.

8. Some titles and analyses of the experimental sections appear to be uninformative. For example, **LLMs Does Not Depend on Observation**, is this a conclusion or a statement? If it is a statement, the authors have already said before that this baseline has no visual input, and it is also contrary to the subsequent titles which are all conclusions. In addition, in the ablation experiments, larger models and more components trained can bring more performance improvements, which is common sense in sense, but the authors put them as ablation experiments alone and do not give any insights.

**Questions:**

1. How to train the reward model $r_{\phi}$? More details are needed.

2. A visual example of task trees is required for better understanding.

---

> ### Author Response · Authors · 2023-11-22
> **Response to Reviewer ABA1 (1/3)**
>
> > The novelty of this paper is limited. Essentially, it uses GPT4 with the prompt engineer to collect data from two embodied environments and then trains a vision-language agent model. Moreover, The agent model does not have a specific framework diagram to show the detailed parts, making it difficult to see which part of the model is its innovation point. I suggest providing a framework diagram to clearly explain where the model is newly proposed in the paper and how it differs from existing methods.
>
> We sincerely thank the reviewer's valuable feedback and we add the diagram in Figure 4. Our core motivation is to develop a domain that remains uncharted to date, i.e., an embodied vision-language model to make reasonable plans and write executable programs. The novel technical part is that (1) it is the first attempt to have a VLM for code writing and (2) it designs a novel solution for planning improvement in the RLEF part, which collects the success/failure information when GPT-4 is collecting the vision-code pairs in the simulator.
>
> > This paper appears to employ an intentional use of uncommon or less frequently used words in many sentences, substituting them for simpler, more common terms that could convey the message more clearly. As a result, the reading experience becomes somewhat disjointed, and the text may come across as rather weird to the reader.
>
> Thank you for the suggestion and we have revised the whole manuscript thoroughly.
>
> > The reason for using both OmniGibson and GTA-V environments to generate data seems not obvious. A more obvious comparison between the two environments is required, such as the visual comparison of the tasks.
>
> The reason of us building OctoVerse is to support the development of our embodied vision-language programmer. OctoVerse is generally based on the existing, most realistic simulators of GTA and OmniGibson, where we add the necessary adaptation to include **the instruction set of callable functions** and **carefully designed tasks**. We include the comparison between OctoVerse with other platforms in appendix C1. The general reason for selecting these two platforms is the consideration of their realistic modeling task designability and scalability. Admittedly, other simulators such as AI2THOR and VirtualHome might also be able to support the Octopus exploration if we also design an instruction set of callable functions and carefully designed tasks, we eventually selected OmniGibson and GTA for realistic modeling and task designability and scalability. The detailed task can be accessed [here](https://filetransfer.io/data-package/WDACj9TO#link) and in the appendix.
>
>
> > In Section Error Management, when the agent executes the wrong command, how does the method perform "error management" on it? It seems that this section only claims the cases under which a task is defined as failure, and does not show how to manage such failure.
>
> Yes, we only annotate commands with "success" or "failure" tags, and these tags are the supervision of the reward modeling in the RLEF part. According to Figure 4 in the paper, all the vision-code pairs in the (a) data collection pipeline will be used for the SFT training in Figure 4(b) Octopus training pipeline, since the GPT-4 does a good job on formatting, meaning that they can write reasonable code. And the "success" or "failure" tag will be used in the RLEF training.
>
> > In Section ENVIRONMENTAL FEEDBACK, what if there are multiple erroneous states in a task sequence? Which are the positive states and which are the error states at this time? And it may not be said that if one of the states is wrong, then its previous states must all be negative.
>
> According to Figure 3, one error state is tagged if, after one step, post-execution states misalign their target states. Alternatively, if a task is not completed as intended, every state within that task is labeled as negative. Sorry for the previous confusion and we have clarified this part in Section 3.3.

---

> > ### Author Response · Authors · 2023-11-22
> > **Response to Reviewer ABA1 (2/3)**
> >
> > > The experimental baselines are unfair and unclear. For Blind LLMs, without visual input to GPT4, GPT4 cannot ground language into the visual environment, which will inevitably lead to worse results. As for TAPA, the reader cannot understand what kind of model it is and its workflow. Even the OVD is not introduced or cited.
> >
> > Sorry for the confusion it made. For Blind LLMs, we provide them with all the environment information in the textual format. Referring to Figure 3, we hope the Blind LLMs could perform as GPT-4 but internalize the system message. For TAPA, we make the implementation details clear in Appendix A, as the TAPA model utilizes the open-vocabulary detection (OVD) technique [Detecting twenty-thousand classes using image-level supervision, ECCV'22] to recognize objects within images. Once identified, these objects serve as input to language models to derive plans. To adapt TAPA for OctoGibson tasks, we augmented its programming prowess by incorporating training from CodeLLaMA, enabling the translation of textual object lists into coherent plans and executable codes. Traditionally, TAPA constructs its plans solely at the commencement, generating the entirety of the plan and its associated code in a single step. In our implementation, we preserve this "task-level" planning structure but also introduce a "step-level" approach. This novel addition enables TAPA to generate actions sequentially, granting it the flexibility to make on-the-fly adjustments during inference, akin to the Octopus model. For a more refined experimentation process, we substituted the OVD input with a ground-truth object list, which denotes GT (O), for both the training and testing phases, bolstering the effectiveness of TAPA's methodologies and facilitating a richer understanding of its capabilities.
> >
> > >  What are the tasks of the four testing environments? The authors did not give a detailed introduction.
> >
> > We are deeply sorry for the unclear explanation in the submission, and revised it in the first part of Section 5.
> > We first set up the OctoGibson to evaluate the performance of Octopus and other related models. Specifically, we are utilizing the metrics of goal task completion score to check whether the task is actually completed in the simulator and the plan score from human evaluation. We totally have 60 evaluation tasks, with 45 from the seen environment, and 15 that are unseen during training. We also have 45 routine tasks and 15 require reasoning. The detailed task can be accessed [here](https://filetransfer.io/data-package/WDACj9TO#link) and in the appendix.
> >
> > > Some titles and analyses of the experimental sections appear to be uninformative. For example,  **LLMs Does Not Depend on Observation**, is this a conclusion or a statement? If it is a statement, the authors have already said before that this baseline has no visual input, and it is also contrary to the subsequent titles which are all conclusions. In addition, in the ablation experiments, larger models and more components trained can bring more performance improvements, which is common sense in sense, but the authors put them as ablation experiments alone and do not give any insights.
> >
> > Thank you for pointing out this we extensively revised the ablation study part, with the conclusion summarized below:
> > - **7B vs. 3B Model Size**: Smaller models like 3B show significantly reduced performance, but the improvement of RLEF is still valid on the smaller model.
> > - **Examining Training Components**: Adjusting both the connector and language decoder in training vision-language models slightly outperforms only adjusting the connector, as shown in Figure 5 (b).
> > - **Significance of Visual Inputs in Task Performance**: Disrupting the sequence of visual inputs in the Octopus model leads to a marked decline in task performance, emphasizing the importance of structured visual inputs, as depicted in Figure 5 (c).
> > - **Performance of GPT-4**: GPT-4, with purely textual input, successfully completes 31 out of 60 tasks, indicating significant potential for further advancements in the field.
> > - **Performance of GPT-4V**: GPT-4V shows promising results in generating code and task execution in image-based environments, though it falls short in specific tasks due to unfamiliarity with the environment and simulator discrepancies, as seen in Figure 5 (e).

---

> > > ### Author Response · Authors · 2023-11-22
> > > **Response to Reviewer ABA1 (3/3)**
> > >
> > > > How to train the reward model. More details are needed.
> > >
> > > We have revised this part in Section 4.3 to make it clear. For reward model configuration, a single-modal CodeLLaMA-7B model is fine-tuned on a specific dataset to serve as a text-only reward model, which evaluates state transitions and helps the agent in determining the most rewarding actions for task execution and completion. It is worth mentioning that we use CodeLLaMA-7B as the reward model since when training the reward model, the input/output pair is only the code and the reward, which is in a pure language scenario. However, the Octopus is a vision-language model and the alignment of the vision and the language part seems to require extra effort to train if we use CodeLLaMA-7B, e.g. more vision-code pair for alignment. Alternatively, we use MPT-7B which is inherited from Otter, where VL alignment is greatly tuned via general vision-language instruct-response pairs. This discussion is also reported in Section 5.1 first paragraph.
> > > For Policy Model Development, the initially supervised fine-tuned model is used as the initial policy model, with another duplicate model trained using Proximal Policy Optimization (PPO) to maximize response rewards. The loss function integrates expected rewards and a Kullback–Leibler (KL) penalty to balance response effectiveness and deviation from the initial policy model.
> > >
> > > >  A visual example of task trees is required for better understanding.
> > >
> > > Thank you for the suggestion! The task list can be accessed [here](https://filetransfer.io/data-package/WDACj9TO#link) and in the supplementary material.

---

> > > > ### Author Response · Authors · 2023-11-23
> > > >
> > > > We sincerely appreciate your great efforts in reviewing this paper. Your constructive advice and valuable comments really help improve our paper. Considering the approaching deadline, please, let us know if you have follow-up concerns. We sincerely hope you can consider our reply in your assessment, and we can further address unclear explanations and remaining concerns if any.
> > > >
> > > > Once more, we are appreciated for the time and effort you've dedicated to our paper.

---

### Official Review · Reviewer_CLdb · 2023-10-30

**Soundness:** 3 good
**Presentation:** 3 good
**Contribution:** 3 good
**Rating:** 5
**Confidence:** 4

**Summary:**

This paper introduced Octopus, a vision-language model mapping the visual input to the action codes. The fine-tuned dataset is collected with the GPT-4 where the simulator feedback is incorporated to generate the system feedback. The author further proposed a RLEF to improve the performance.

**Strengths:**

- This paper proposed a novel VLM  to transfer the visual input to the executable codes, driving the agents.
- The GPT-4 along with a simulator is used to collect training datasets. OminiGibson and OctoGTA are used respectively.
- An RLEF module is used to boost the model's performance further.

**Weaknesses:**

- More related works in Section 2.1 are needed to help the reviewer identify the paper's contributions, like [1-4].
- The authors term the simulation they used "OctoGibson" which is built upon OmniGibson. Can the authors give more details to elaborate the main difference between them, or did they just use that Simulator to collect the dataset?
- A better format is needed. Some lines need a reformat in the revised version. One example is "3.2 Instructions From Exploration", lines above and below seem to belong to the same paragraph.
- More experiments are needed. The author only conducts the experiments on the datasets they collected and lacks a direct comparison with more relative frameworks as discussed in Section 2.1.

[1] Huang, Wenlong, et al. "Voxposer: Composable 3d value maps for robotic manipulation with language models." arXiv preprint arXiv:2307.05973 (2023).
[2] Lin, Kevin, et al. "Text2motion: From natural language instructions to feasible plans." arXiv preprint arXiv:2303.12153 (2023).
[3] Huang, Siyuan, et al. "Instruct2Act: Mapping Multi-modality Instructions to Robotic Actions with Large Language Model." arXiv preprint arXiv:2305.11176 (2023).
[4] Yu, Wenhao, et al. "Language to Rewards for Robotic Skill Synthesis." arXiv preprint arXiv:2306.08647 (2023).

**Questions:**

- The author uses the GPT-4 to collect the training data, and one implicit assumption is that the performance of the GPT-4 is optimal or near-optimal. A comparison between the GPT-4 generated data sample and the human-collected sample would help. Or, did the author conduct some data quality control before the use of dataset?

- When using GPT-4 plus a simulator to collect the dataset, the location of the target object is directly obtained from the simulator? And this information be stored and used for later training? With this approach, the final complete robotic system still needs a separate vision model besides the ViT-L in the VLM. Can the author give some discussion on this design choice?

- RLEF. It is very interesting to see the usage and effectiveness of RLEF. However, I am curious as to why you chose CodeLLaMA-7B as the reward model while using MPT-7B for the complete VLM?

- In Table 2, there is a comparison between Octopus and  MPT-7B. Also, the performance is not consistently superior, a further discussion is needed. And the metrics' definition is needed to help the understanding.

- Ablation: 3B: what is the 3B model?

- The author inputs 10 images to the VLM and discusses the standard version vs the random version.  Would other designs help?

- The author states multiple times with "open-sourcing" in the main text, a link to the anonymous website would be helpful.

- See weakness.

---

> ### Author Response · Authors · 2023-11-22
> **Response to Reviewer CLdb (1/2)**
>
> > More related works in Section 2.1 are needed to help the reviewer identify the paper's contributions, like [1-4].
>
> We sincerely thank the reviewer for pointing out these important works, by comparing them we can better show the accurate standing of the Octopus work in the community. We add all these four works in the Table 1, and we show that the proposed Octopus distinguishes itself from other models as an end-to-end, unified vision-language model for both plan and code generation.
>
> > The authors term the simulation they used "OctoGibson" which is built upon OmniGibson. Can the authors give more details to elaborate on the main difference between them, or did they just use that Simulator to collect the dataset?
>
> The OctoVerse is generally based on the existing realistic simulators of GTA and OmniGibson, where we add the necessary adaptation to include **the instruction set of callable functions** and **carefully designed tasks** that make the development of the embodied vision-language programmer training possible.
>
> >  A better format is needed. Some lines need a reformat in the revised version. One example is "3.2 Instructions From Exploration", lines above and below seem to belong to the same paragraph.
>
> Thank you for pointing it out! Yes, we have drastically revised the whole draft several rounds and made sure it is easy to understand with good structure.
>
> > More experiments are needed. The author only conducts the experiments on the datasets they collected and lacks a direct comparison with more relative frameworks as discussed in Section 2.1.
>
> Yes, we have added several experiments in Section 5.1 and Table 2, where we add throughout comparison with "Blind LLM" including LLaMA, CodeLLaMA, TAPA, which can only "listen" to what is inside the environment. We also implement the EmbodiedGPT for comparison. These experiments have shown several conclusions including:
> - Blind LLMs Struggle with Extended Input Content.
> - CodeLLaMA Improves Coding but not Planning.
> - Octopus Demonstrates Superior Task Generalization Ability as it integrates the visual input.
> -  RLEF Enhances Octopus’s Planning Strategy, which is more significant than the components in EmbodiedGPT.
>
> | Model             | Vision Model | Language Model | Seen Env | Unseen Env | Follow | Reason | All   |
> |------------------|--------------|----------------|----------|------------|--------|--------|-------|
> | LLaMA            | GT (O+R)     | LLaMA2-7B      | 0.07 / 0.11 | 0.13 / 0.13 | 0.11 / 0.16 | 0.00 / 0.00 | 0.08 / 0.12 |
> | CodeLLaMA        | GT (O+R)     | CodeLLaMA-7B   | 0.09 / 0.20 | 0.20 / 0.40 | 0.16 / 0.31 | 0.00 / 0.07 | 0.12 / 0.25 |
> | TAPA (task-level) | ~~OVD~~ GT (O) | CodeLLaMA-7B   | 0.09 / 0.36 | 0.13 / 0.33 | 0.11 / 0.36 | 0.06 / 0.33 | 0.10 / 0.35 |
> | TAPA (step-level) | ~~OVD~~ GT (O) | CodeLLaMA-7B   | **0.16 / 0.42** | 0.13 / 0.27 | **0.18 / 0.38** | 0.07 / 0.40 | 0.15 / 0.38 |
> | EmbodiedGPT       | CLIP-ViT     | MPT-7B         | 0.04 / 0.36 | 0.27 / **0.53** | 0.13 / 0.38 | 0.00 / 0.40 | 0.10 / 0.40 |
> | Octopus (SFT Only) | CLIP-ViT   | MPT-7B         | 0.11 / 0.33 | 0.27 / 0.47 | 0.16 / 0.38 | 0.13 / 0.33 | 0.15 / 0.37 |
> | Octopus (SFT + RLEF) | CLIP-ViT | MPT-7B         | 0.13 / 0.38 | **0.33 / 0.53** | **0.18 / 0.40** | **0.20 / 0.53** | **0.18 / 0.42** |
>
> These experimental results show the promising direction of the end-to-end vision language models for planning and programming.
>
> > The author uses the GPT-4 to collect the training data, and one implicit assumption is that the performance of the GPT-4 is optimal or near-optimal. A comparison between the GPT-4 generated data sample and the human-collected sample would help. Or, did the author conduct some data quality control before the use of dataset?
>
> Thank you for the great question! Actually, when we use GPT-4 for the automatic data collection pipeline, the "environmental feedback" generally acts like the data quality controller. More specifically, when GPT-4 writes the plan, code, and target state, the code will be run in the simulator and check whether the target state is met and whether the final mission is satisfied. All these results of success/failure will be valuable resources for RLEF training. In fact, we intensively designed a very comprehensive system message for GPT-4 and it is good at following the structure with near-perfect code format (seldom syntax error occurs), and the syntax correctness can be identified by the simulator execution and we will remove these with syntax incorrectness when training SFT.

---

> > ### Author Response · Authors · 2023-11-22
> > **Response to Reviewer CLdb (2/2)**
> >
> > > RLEF. It is very interesting to see the usage and effectiveness of RLEF. However, I am curious as to why you chose CodeLLaMA-7B as the reward model while using MPT-7B for the complete VLM?
> >
> > Thank you for pointing it out! Actually, we use CodeLLaMA-7B as the reward model since when training the reward model, the input/output pair is only the code and the reward, which is in a pure language scenario. However, the Octopus is a vision-language model and the alignment of the vision and the language part seems to require extra effort to train if we use CodeLLaMA-7B, e.g. more vision-code pair for alignment. Alternatively, we use MPT-7B which is inherited from Otter, where VL alignment is greatly tuned via general vision-language instruct-response pairs. This discussion is also reported in Section 5.1 first paragraph.
> >
> > > In Table 2, there is a comparison between Octopus and MPT-7B. Also, the performance is not consistently superior, a further discussion is needed. And the metrics' definition is needed to help the understanding.
> >
> > In Table 2, Octopus reaches a good result on almost all the metrics except the seen environment, showing that Octopus, with the vision as the input can have a better generalization ability in planning. The pure language models, when given the familiar textual input that reflects the environment, seem to have a better result. But we believe task and environmental generalization is a more important aspect. For the detailed explanation of metrics, we fix this part at the beginning of section 5: We first set up the OctoGibson to evaluate the performance of Octopus and other related models. Specifically, we are utilizing the metrics of goal task completion score to check whether the task is actually completed in the simulator and the plan score from human evaluation. We totally have 60 evaluation tasks, with 45 from the seen environment, and 15 that are unseen during training. We also have 45 routine tasks and 15 require reasoning. Please note that models like Octopus might not always accurately identify specific object names as they appear in the simulator (e.g., ``water\_bottle\_189''). To address this, we implement a post-processing step for the generated code, substituting generic object references with their exact names from the simulator with simple string similarity matching.
> >
> > >  Ablation: 3B: what is the 3B model?
> >
> > 3B model compares to the 7B model with respect to the language model size. We clarify this part in the Figure 5 caption. We show that the RLEF component is effective on both 3B and 7B models.
> >
> > > The author inputs 10 images to the VLM and discusses the standard version vs the random version. Would other designs help?
> >
> > Yes, it is valuable to try other numbers of image inputs for VLM training and we are still doing the experiment. The main purpose of the discussion on standard version and random version is to test the necessity of the vision input and it shows the vision input matters.
> >
> > > The author states multiple times with "open-sourcing" in the main text, a link to the anonymous website would be helpful.
> >
> > Thank you for pointing it out! We add the open-source code in the appendix.

---

> > > ### Author Response · Authors · 2023-11-23
> > >
> > > We sincerely appreciate your great efforts in reviewing this paper. Your constructive advice and valuable comments really help improve our paper. Considering the approaching deadline, please, let us know if you have follow-up concerns. We sincerely hope you can consider our reply in your assessment, and we can further address unclear explanations and remaining concerns if any.
> > >
> > > Once more, we are appreciated for the time and effort you've dedicated to our paper.

---

### Official Review · Reviewer_qA1p · 2023-11-01

**Soundness:** 1 poor
**Presentation:** 1 poor
**Contribution:** 2 fair
**Rating:** 3
**Confidence:** 4

**Summary:**

The manuscript proposes a model and simulator for instruction-following tasks in Embodied AI, leveraging GPT-4 for a human-model-agent task-execution paradigm.

**Strengths:**

The manuscript makes reference to relevant methodology in EAI — designing agents that include foundation models, which perform intermediate reasoning tasks

**Weaknesses:**

Section 1 / Throughout — The manuscript forgets to properly motivate its contributions. What problem is this work supposed to be solving? What research questions are examined by this manuscript?

Section 1 — Most of the Introduction section is unnecessary. The space should instead be used to describe what is added on top of GPT-4 to make the model proposed in this paper a sufficiently distinct contribution. How is OctoVerse different from other EAI simulators? Why does the community need OctoVerse? What problems can be solved in OctoVerse that cannot be solved elsewhere? The manuscript fails both to motivate and explicitly describe its contributions.

Section 2 — Call it “Related Work”. The dimensions on which this section compares the proposed work with the prior art are all wrong. Firstly, because the manuscript is attempting to propose a new environment and tasks, it should identify the limitation of other, similar simulators/datasets and explicitly discuss the proposed improvements. Regarding claims for novel modeling contributions, the manuscript must first propose research questions or problems that the approach attempts to solve. Next, a set of related work can be organized to discuss their attempts at answering said research question and solving said problems, as well as discuss their limitations or weaknesses. Finally, this structure affords the manuscript to describe how its proposed work improves on the prior art, according to those research questions and identified problem(s).

Section 3 — The manuscript does not make clear what was originally provided by OmniGibson / GTA-V, versus what is added by OctoVerse. Also, again, the manuscript is missing motivation for why anyone should use its proposed environment. The problem formulation needs a lot of work.

**Questions:**

N/A — see above.

---

> ### Author Response · Authors · 2023-11-22
> **Response to Reviewer qA1p (1/2)**
>
> > Section 1 / Throughout — The manuscript forgets to properly motivate its contributions. What problem is this work supposed to be solving? What research questions are examined by this manuscript?
>
> We sincerely appreciate the constructive feedback regarding the need to more clearly articulate the motivation behind our work. We have thoroughly revised our Introduction section (highlighted in the submission paper).
> Our core motivation is to develop a domain that remains uncharted to date, i.e.,**an embodied vision-language models to make reasonable plans and write executable programs”**. While there are existing models like ToolFormer, HuggingGPT, ViperGPT, and VisProg that delve into programming with non-visual inputs, Octopus stands out in its endeavor to merge these programming models with visual stimuli. Consequently, recognizing the lack of suitable environments for conducting such innovative experiments, we have developed OctoGTA and OctoGibson, enabling us to explore this new territory of vision-language model programming. We hope this explanation can help clarify the standing of the motivation and the purpose of the two environments.
>
> > Section 1 — Most of the Introduction section is unnecessary. The space should instead be used to describe what is added on top of GPT-4 to make the model proposed in this paper a sufficiently distinct contribution. How is OctoVerse different from other EAI simulators? Why does the community need OctoVerse? What problems can be solved in OctoVerse that cannot be solved elsewhere? The manuscript fails both to motivate and explicitly describe its contributions.
>
> Thank you again for the constructive suggestions! We are generally using GPT-4 to prepare for the vision input and executable program pairs, and the core contribution is to illustrate the embodied vision-language models can have better achievement than the unimodal language models, with experimental support in Table 2, where we see (1) the drawbacks of unimodal LLMs: they struggle with extended input content that describes the visual input; and (2) The significance of visual inputs in task performance. The OctoVerse is generally based on the existing, most realistic simulators of GTA and OmniGibson, where we add the necessary adaptation to include **the instruction set of callable functions** and **carefully designed tasks**. We include the comparison between OctoVerse with other platforms in Appendix C1. The general reason for selecting these two platforms is the consideration of their realistic modeling task designability and scalability. Admittedly, other simulators such as AI2THOR and VirtualHome might also be able to support the Octopus exploration if we also design an instruction set of callable functions and carefully designed tasks, we eventually selected OmniGibson and GTA for realistic modeling and task designability and scalability. In the revised Appendix C1, we incorporate various attributes that enhance diversity and realism, noting a substantial advancement in both OctoGTA and OctoGibson simulation environments. First, a wide range of well-formulated tasks are established in Octopus's simulator showcasing from fine-grained indoor routine activities to outdoor open-world tasks. This contrasts with other simulation environments that often target a relatively restricted set of activities.
>
> | Simulation Environment | Kinematics | Continuous Extended States | Flexible Materials | Deformable Bodies | Realistic Fluid | Realistic Action Execution | Task Planning and/or Control | Game-Based or World-Based | Well-Formulated Tasks | Code Execution |
> |------------------------|------------|----------------------------|--------------------|-------------------|-----------------|---------------------------|-------------------------------|---------------------------|------------------------|----------------|
> | OpenAIGym | ✓ | ✕ | ✕ | ✕ | ✕ | ✓ | C | G | ✕ | ✓ |
> | Matterport3D | ✕ | ✕ | ✕ | ✕ | ✕ | ✕ | C | W | ✕ | ✕ |
> | AI2THOR | ✓ | ✕ | ✕ | ✕ | ✕ | ✕ | TP | G | ✕ | ✓ |
> | VirtualHome | ✕ | ✕ | ✕ | ✕ | ✕ | ✕ | TP | G | ✕ | ✕ |
> | House3D | ✕ | ✕ | ✕ | ✕ | ✕ | ✕ | TP | W | ✕ | ✕ |
> | Habitat 1.0 | ✓ | ✕ | ✕ | ✕ | ✕ | ✓ | C | W | ✕ | ✓ |
> | Robosuite | ✓ | ✕ | ✕ | ✕ | ✕ | ✓ | C | W | ✕ | ✓ |
> | RFUniverse | ✓ | ✕ | ✓ | ✓ | ✓ | ✓ | TP+C | W | ✕ | ✓ |
> | Minecraft | ✓ | ✕ | ✓ | ✕ | ✕ | ✓ | TP+C | G | ✓ | ✓ |
> | GTA | ✓ | ✓ | ✓ | ✓ | ✓ | ✓ | TP+C | G | ✕ | ✕ |
> | Omnigibson | ✓ | ✓ | ✓ | ✓ | ✓ | ✓ | TP+C | W | ✕ | ✕ |
> | OctoGTA | ✓ | ✓ | ✓ | ✓ | ✓ | ✓ | TP+C | G | ✓ | ✓ |
> | Octogibson | ✓ | ✓ | ✓ | ✓ | ✓ | ✓ | TP+C | W | ✓ | ✓ |

---

> > ### Author Response · Authors · 2023-11-22
> > **Response to Reviewer qA1p (2/2)**
> >
> > > Section 2 — Call it “Related Work”. The dimensions on which this section compares the proposed work with the prior art are all wrong. Firstly, because the manuscript is attempting to propose a new environment and tasks, it should identify the limitation of other, similar simulators/datasets and explicitly discuss the proposed improvements. Regarding claims for novel modeling contributions, the manuscript must first propose research questions or problems that the approach attempts to solve. Next, a set of related work can be organized to discuss their attempts at answering said research question and solving said problems, as well as discuss their limitations or weaknesses. Finally, this structure affords the manuscript to describe how its proposed work improves on the prior art, according to those research questions and identified problem(s).
> >
> > We are sorry for the initial unclear and misleading writing on our motivation and goal. Actually, **the goal of the paper is to introduce a novel "vision-language programming model" that can write executable code to complete tasks according to the visual environment**, and the proposed simulators are generally based on the existing platform to achieve the vision-language programmer. In this sense, in related work, we are comparing different models and highlighting that the proposed Octopus distinguishes itself from other models as a unified vision-language model for both plan and code generation.

---

> > > ### Author Response · Authors · 2023-11-23
> > >
> > > We sincerely appreciate your great efforts in reviewing this paper. Your constructive advice and valuable comments really help improve our paper. Considering the approaching deadline, please, let us know if you have follow-up concerns. We sincerely hope you can consider our reply in your assessment, and we can further address unclear explanations and remaining concerns if any.
> > >
> > > Once more, we are appreciated for the time and effort you've dedicated to our paper.

---

### Meta-Review · Area_Chair_PfdA · 2023-12-09

**Metareview:**

The paper presents an embodied vision-language planner and an environment to evaluate it.

All reviewers find the paper premature for publications -- confusing motivation and lack of clarity; mistakes in the manuscript; overall insufficient writing quality. Further, some of the reviewers find lack of novelty and lack of empirical justification of the method.

**Justification For Why Not Higher Score:**

The paper received two reject (3) and two borderline reject (5) ratings. All reviewers find the paper premature for publications -- confusing motivation and lack of clarity; mistakes in the manuscript; overall insufficient writing quality. Further, some of the reviewers find lack of novelty and lack of empirical justification of the method. Hence, unfortunately the paper is rejected from ICLR 2024.

**Justification For Why Not Lower Score:**

N/A

---

### Decision · Program_Chairs · 2024-01-16

Reject